# Reliability Analysis of the Main Pier during the Construction Period of HLCR Bridges

Ningning Qi [1], Baosheng Xu [1], Tianjing Zhang [2],* and Qingfu Li [2]

1   China Construction Seventh Engineering Division Corp. Ltd., Zhengzhou 450004, China; qiningdux@126.com (N.Q.); jtgsxbs@126.com (B.X.)
2   School of Water Conservancy Engineering, Zhengzhou University, Zhengzhou 450001, China; lqflch@zzu.edu.cn
*   Correspondence: 202012222024294@gs.zzu.edu.cn; Tel.: +86-178-0386-3022

**Abstract:** In the process of cradle construction for high-pier, long-span, continuous, rigid-frame (HLCR) bridges, the strength failure and instability damage of the main pier will directly affect the bridge construction safety. Therefore, it is necessary to study the reliability of the main pier during the construction of HLCR bridges. This paper starts from the factors that easily affect the stability of the main pier during HLCR bridge cradle construction, establishes the resistance and load probability model of the main pier during the maximum cantilever stage and the maximum unbalanced load of the continuous, rigid-frame bridge's hanging cradles, fully considers the influence of random factors on the reliability of the pier, and calculates and analyzes the reliability index β through calculation examples. The results show that the changes of various random factors during the construction process have different degrees of influence on the reliability of the bridge pier. Our work provides a basis for the safe control of hanging cradles in the construction of HLCR bridges.

**Keywords:** cradle construction; reliability; pier; probability model; JC method

## 1. Introduction

In recent years, the construction of China's infrastructure has developed rapidly. According to incomplete statistics from relevant departments, the number of bridges in service in China exceeded 800,000 by the end of 2020 [1]. In just 20 years of the 21st century, China's bridge construction has stepped into the "beyond" stage with a steady pace, opening a new era of "super bridge" construction, and the level of bridge construction is constantly hitting the worldwide maximum height. In promoting poverty alleviation work in China, the importance of remote mountainous areas to transportation construction efforts has also increased. In 1953, the former Federal Republic of Germany successfully built the main span of the 114.2 m Worms Bridge, which was the first cantilever bridge in the world to have a rigid structure [2]. The first continuous rigid bridge built in China was the Luoxi Bridge in Guangzhou, built in 1984 with a main span of 180 m. It became known as the "Luoxi Flying Rainbow" and was the first bridge of its kind in Asia and the sixth in the world when it was built [3]. Today, with the increasing scale of bridge construction, the technical difficulty of construction and management is also increasing, which brings the disadvantage that the accident rate of bridges is getting higher. This is not only causing huge economic loss but also has an extremely negative social impact. The construction period is comparably riskier than the design and use periods, and the reliability of the project during the construction period affects the safety and durability of the structure after it is put into operation. To ensure that the bridge construction can meet the design requirements and the relevant specification standards, it is necessary to conduct a dynamic reliability analysis for each construction link during the construction period. However, at present, China does not pay sufficient attention to studying the reliability of bridge structures during the construction period, which leads to a low safety and quality of

structures during the construction period, and as a result, the frequent occurrence of bridge construction quality errors and safety accidents, with serious economic losses and huge social impacts.

To date, there has been some research on the reliability of bridge structures during the construction period at home and abroad. There are roughly three evolutionary stages in the research: the first stage is mainly based on the subjective judgment of experts on whether the structure will collapse, break or other accidents will occur based on their experience and after observing the structure. The second stage is that people judge the reliability of the structure based on the theory of the safety factor. In the third stage, reliability is used to express the probability that the structure will achieve the expected goal. The reliability theory is based on the probability theory and evaluates and optimizes the safety, quality and reliability of the structure through mathematical calculation, taking into account the uncertainty and randomness of all important parameters that affect the safety and reliability of the structure during the design and construction process.

Internationally, the boom of structural reliability theory development was in the middle of the 20th century, in which the classical reliability theory was born, became accepted and started to be used by the engineering community after in-depth research. In China, the reliability theory developed rapidly in the decade from 1984 to 1994. Dalian University of Technology and Tsinghua University have done a lot of work on reliability-based structural durability research [4]. Analysis of the factors affecting the quality and safety of bridges at each stage of construction, how to express these factors and how to quantify them has become one of the main tasks of bridge reliability research during the construction period. Since HLCR bridges are mostly constructed in mountainous areas, and the cantilever hanging cradle method is most used to construct the main girders of a continuous rigid bridge, the overall instability of the bridge will reach a maximum when the construction reaches the maximum cantilever state, which seriously affects the safety of the bridge during the construction period.

In recent years, research on the reliability of the overturning resistance of the maximum cantilever construction stage of a continuous rigid bridge has been increasing. Casas [5] found that when using the balanced cantilever method to build concrete bridges, it is easy to lose the overall stability due to structural overturning. From the perspective of safety, the magnitude of cantilever construction force is studied to achieve a similar level of reliability across the span of these bridges. Sexsmith [6] analyzed the safety problems of temporary works during construction. The reliability level was comprehensively selected according to the exposure time of the temporary load, construction cost of temporary works and other factors. Zhang et al. [7] took the Jingzhou Yangtze River Highway Bridge as the research object, and carried out a detailed calculation and analysis of its overall stability and parameter sensitivity in the most unstable state during the construction phase. The research results showed that the self-weight error of the superstructure on both sides of the pier during cantilever construction was the main factor affecting the overturning stability of the bridge during the construction period, while the wind load had little effect on the stability reliability, which provided a subsequent bridge overturning resistance research reference. Zhang [8] calculated the reliability of two failure modes when the cantilever cast-in-place method was used to construct up to the last section of the girder block, and performed a parametric sensitivity analysis. The results of the study provide a reference for managing the construction phase of the structure.

Hedjazi et al. [9] used ABAQUS software to establish a three-dimensional finite element model for cantilever balanced construction of segmental bridges, and tracked the structural response during construction and the whole service life. Significant time-dependent effects on bridge deflection and internal forces and stress redistribution were observed. Based on the estimated distribution of the dead load, live load and wind load, Catbas et al. [10] conducted reliability evaluation research on the main truss members and the whole structural system of the longest along-span truss bridge in the United States. The results show that the response to temperature significantly impacts the reliability of the

whole system. Sun [11] analyzed the factors affecting the alignment of the main girders during the bridge construction phase of the Zhenyuan Canal Bridge, and proposed that the deflection change of the girder end should be controlled at each step of the bridge construction to obtain a reasonable alignment of the completed bridge. After analyzing the overturning stability of a three-span continuous girder bridge under the most unfavorable conditions in the construction phase, Zorong et al. [12] identified the parameters with the greatest influence on the reliability index and determined their value ranges, thus improving the safety of the solid structure during construction. By considering the actual bearing capacity and actual situation of the bridge, Vican et al. [13] proposed a complex existing bridge evaluation method based on the defect point method. Using the developed method, the priority of a bridge repair can be determined. Cheng [14] analyzed and studied the stability reliability of an HLCR bridge when the cantilever was constructed to the longest state, and concluded that the elastic modulus and self-weight of concrete had a large effect on the stability reliability of the bridge, while the wind load had little effect.

At present, there are many methods to calculate the structural reliability index; suitable analysis methods can be selected for the characteristics of different engineering structures, subject to the continuous improvement of our knowledge by domestic and foreign research scholars [15]. The JC method, MC method, and response surface method are the most commonly used and widely studied reliability calculation methods at present. The earliest international method to calculate reliability was the centroid method proposed by Conell [16] in the early stage of structural reliability research, which is also the simplest method. The basic idea is to first make a Taylor series expansion of the nonlinear function at the centroid of the mean of the random variable and retain it up to the primary term, and then approximate the mean and standard deviation of the function [17]. It is suitable to calculate the reliability when the distribution type of the random variable is difficult to determine but its first- and second-order moments are known. In 1951, the response surface method was born [18]. The basic idea of this is to use an explicit and easy-to-handle function to fit the complex and nonlinear implicit function in engineering structures. Then, on the premise of ensuring a certain calculation accuracy, the calculation amount is significantly reduced [19]. This method is mainly aimed at functions with a high degree of nonlinearity, large degree of fluctuation of the geometric surface of the limit state equation, low iterative accuracy, unstable iterative process and difficult convergence. Wong [20] proposed to apply the response surface method to the solution of reliability, and Bucherg and Bou [21] to engineering reliability. In 1974, Hasofer and colleagues proposed the checkpoint method (also called the JC method) [22]. The JC method was developed with the first second-order moment theory. The JC method considers the actual distribution of random variables in the function and takes the checkpoint on the failure surface, which is consistent with the geometric meaning of the maximum failure probability of the structure, thereby improving the accuracy of the reliability calculation [23]. Meanwhile, the Monte-Carlo (MC) method (referred to as the MC method) directly applies the principle of numerical statistics to a large number of random samples of the structure, analyzes the resulting values to determine whether they will lead to structural failure, and introduces the probability of failure or structural reliability of the structure based on a large number of sample values [24]. The accuracy of the MC method calculation results depends only on the selection and number of samples. The MC method is widely used in engineering because it does not require the establishment of complex mathematical formulas. In recent years, domestic and foreign research scholars have focused their research on the MC method, on the selection of sampling methods and on how to reduce the number of simulations while reducing the calculation error. Based on this, they have proposed the significant sampling [25], stratified sampling [26], control variable [27] and pairwise variables methods [28].

This study is based on the construction of the hanging cradles of a cross-river bridge project, and it takes "the reliability of hanging baskets of an HLCR bridge" as the research content. This research profoundly analyzes the current situation of domestic and foreign research, and it closely combines the construction method and process characteristics of the

main girder part of a long-span, continuous, rigid-frame bridge. The JC method is used to establish a reliability analysis of the construction period and an evaluation model, which provide a technical basis for ensuring the construction safety and quality of the structure.

## 2. Theoretical Foundation

### 2.1. Checkpoint Method (JC Method)

Let the function of the structure be $Z = g(x_1, x_2, \cdots, x_n)$, assuming point $P(x_1^*, x_2^*, \cdots x_n^*)$ is the design checkpoint on the limit state surface, and satisfying [29]

$$Z = g(x_1^*, x_2^*, \cdots x_n^*) = 0. \tag{1}$$

Here, the underlying variables are assumed to obey a normal distribution where the mean is $\mu = (\mu_{x_1}, \mu_{x_2}, \cdots, \mu_{x_n})$, and the standard deviation is $\sigma = (\sigma_{x_1}, \sigma_{x_2}, \cdots, \sigma_{x_n})$.

Expanding the function $Z$ into a Taylor series at point $P$ and retaining the primary term, the mean value of $Z$ is

$$\mu_Z = g(x_1^*, x_2^*, \cdots x_n^*) + \sum_{i=1}^{n} (\mu_{x_i} - x_i^*) \left. \frac{\partial g}{\partial x_i} \right|_{x^*} \tag{2}$$

Since the design checkpoint is on the failure boundary, $g(x_1^*, x_2^*, \cdots x_n^*) = 0$, so

$$\mu_Z \approx \sum_{i=1}^{n} (\mu_{x_i} - x_i^*) \left. \frac{\partial g}{\partial x_i} \right|_{x^*} \tag{3}$$

$$Z \approx g(x_1^*, x_2^*, \cdots x_n^*) + \sum_{i=1}^{n} (x_i - x_i^*) \left. \frac{\partial g}{\partial x_i} \right|_{x^*} \tag{4}$$

Considering the correlation of random variables, then the variance of $Z$ can be solved with

$$\sigma_Z^2 \approx \sum_{i=1}^{n} \sum_{j=1}^{n} \left( \left. \frac{\partial g}{\partial x_i} \right|_{x^*} \left. \frac{\partial g}{\partial x_j} \right|_{x^*} \right) Cov(x_i, x_j) = \sum_{i=1}^{n} \sum_{j=1}^{n} \left( \left. \frac{\partial g}{\partial x_i} \right|_{x^*} \left. \frac{\partial g}{\partial x_j} \right|_{x^*} \right) \rho_{x_i x_j} \sigma_{x_i} \sigma_{x_j} \tag{5}$$

$$\sigma_Z = \sqrt{\sum_{i=1}^{n} \sum_{j=1}^{n} \left( \left. \frac{\partial g}{\partial x_i} \right|_{x^*} \left. \frac{\partial g}{\partial x_j} \right|_{x^*} \right) \rho_{x_i x_j} \sigma_{x_i} \sigma_{x_j}} = \sum_{i=1}^{n} \alpha_i \left. \frac{\partial g}{\partial x_i} \right|_{x^*} \sigma_{x_i} \tag{6}$$

$$\alpha_i = \frac{\sum\limits_{j=1}^{n} \left. \frac{\partial g}{\partial x_j} \right|_{x^*} \rho_{x_i x_j} \sigma_{x_j}}{\sqrt{\sum\limits_{i=1}^{n} \sum\limits_{j=1}^{n} \left( \left. \frac{\partial g}{\partial x_i} \right|_{x^*} \left. \frac{\partial g}{\partial x_j} \right|_{x^*} \right) \rho_{x_i x_j} \sigma_{x_i} \sigma_{x_j}}} \tag{7}$$

Here, $\alpha_i$ is the sensitivity coefficient, which indicates the relative effect of the $i$-th random variable on the standard deviation.

$$\beta = \frac{\mu_Z}{\sigma_Z} = \frac{g(x_1^*, x_2^*, \cdots x_n^*) + \sum\limits_{i=1}^{n} \frac{\partial g}{\partial x_i} (\mu_{x_i} - x_i^*) |_{x^*}}{\sum\limits_{i=1}^{n} \left( \alpha_i \sigma_{x_i} \cdot \left. \frac{\partial g}{\partial x_i} \right|_{x^*} \right)} \tag{8}$$

Multiplying $\beta$ into the denominator and sorting gives:

$$\sum_{i=1}^{n} \left. \frac{\partial g}{\partial x_i} \right|_{x^*} (\mu_{x_i} - x_i^* - \beta \alpha_i \sigma_{x_i}) = 0 \tag{9}$$

As $\left.\frac{\partial g}{\partial x_i}\right|_{x^*} \neq 0$, there must be $\mu_{x_i} - x_i^* - \beta\alpha_i\sigma_{x_i} = 0 (i = 1, 2, \cdots, n)$, i.e.,

$$x_i^* = \mu_{x_i} - \beta\alpha_i\sigma_{x_i} \tag{10}$$

$$g(x_1^*, x_2^*, \cdots x_n^*) = 0 \tag{11}$$

In a special case, when the random variables are independent of each other, then Equation (6) can be changed to:

$$\sigma_Z = \sqrt{\sum_{i=1}^{n}\left(\sigma_{x_i}\left.\frac{\partial g}{\partial x_i}\right|_{x^*}\right)^2} = \sum_{i=1}^{n}\alpha_i'\sigma_{x_i}\left.\frac{\partial g}{\partial x_i}\right|_{x^*} \tag{12}$$

$$\alpha' = \frac{\sigma_{x_j}\left.\frac{\partial g}{\partial x_j}\right|_{x^*}}{\sqrt{\sum_{i=1}^{n}\left(\sigma_{x_i}\left.\frac{\partial g}{\partial x_i}\right|_{x^*}\right)^2}} \tag{13}$$

The iterative method is commonly used to solve the design verification point $P^*\left(x_1^*, x_2^*, \cdots x_n^*\right)$ and the corresponding reliability index $\beta$. The iterative procedure is as follows:

(1) Assume that the design checkpoint $P^*\left(x_1^*, x_2^*, \cdots x_n^*\right)$ is initially assigned (generally taken as the mean value point);

(2) Carry out equivalent normalization of the non-normal variable $x_i$. The statistical parameters $\mu_{y_i}$ and $\sigma_{y_i}$ of its equivalent normal distribution $y_i$ are calculated and used in place of $\mu_{x_i}$ and $\sigma_{x_i}$. This is recorded as $\mu_{x_i} = \mu_{y_i}, \sigma_{x_i} = \sigma_{y_i} (i = 1, \cdots, n)$;

(3) From Equation (7), calculate the value of $\alpha_i$, including $\left.\frac{\partial g}{\partial x_i}\right|_{x^*}$ and $\left.\frac{\partial g}{\partial x_j}\right|_{x^*}$;

(4) Using Equation (8), calculate the value of $\beta$;

(5) Using Equation (10), calculate the new value of $x_i^*$;

(6) Repeat (2)–(5) with this $x_i^*$ until the difference between the first and second $\|x^*\|$ or the absolute value of $\beta$ is less than the allowable error $\varepsilon$.

## 2.2. Analysis of Indeterminate Factors Affecting the Resistance of the Main Pier in Cradle Construction

Research by domestic and foreign scholars shows that there are many factors affecting the resistance of bridge structures, and the lack of resistance of structural members during the construction period is an important cause of bridge structure failure. Therefore, when conducting a reliability analysis of the main pier during the construction period of the bridge structure, the factors affecting the resistance of the pier should be reasonably analyzed first.

Factors affecting the structural resistance $R$ of bridges are usually considered from the following three aspects: the uncertainty of material properties $K_M$, the uncertainty of geometric parameters $K_A$ and the uncertainty of the computational model $K_P$ [30]. Since these factors are generally random variables, the resistance of a structure or member is often a function of multiple random variables. In determining the statistical characteristics of the resisting force, the statistical parameters of each influencing factor are usually determined first, and then the statistical parameters and probability distribution types of the resisting force are deduced according to the functional relationship between the resisting force and the relevant factors. According to the results of a large number of studies, the uncertainty of geometric parameters in practical engineering is small [1], so its variation has little influence on the reliability index. For the convenience of calculation, this paper mainly considers the influence of the uncertainty of material properties $K_M$ and calculation mode $K_P$ on the uncertainty of the pier's body resistance.

The uncertainty of the material properties of structural members mainly refers to the variability of the material properties in the members due to the quality of the material and

the fabrication process caused by the loading condition, shape and size, environmental conditions and other factors. This uncertainty can be expressed in $K_M$ [30]:

$$K_M = \frac{f_{st}}{\omega_0 f_k} \tag{14}$$

where $f_{st}$ represents the actual material property values in the structural elements, $f_k$ the standard value of the material properties of the specimen as specified in the specification and $\frac{1}{\omega_0}$ is the coefficient reflecting the difference between the material properties of the structural elements and the material properties of the specimens.

The uncertainty of the component resistance calculation model mainly refers to the variability caused by the approximation of certain basic assumptions made in the process of resistance calculation and by the imprecision of the calculation formula, which can be expressed by the random variable $K_P$ [30]:

$$K_P = \frac{R_s}{R_j} \tag{15}$$

where $R_s$ is the actual resistance value of the component, which can generally be measured in the test or accurately calculated; $R_j$ is the resistance value calculated according to the specification formula, and the measured values of material properties and geometric dimensions are used for calculation.

The statistical analysis of $K_M$ and $K_P$ leads to the mean value $\mu$ with the coefficient of variation $\delta$. The commonly used statistical parameters for the uncertainty $K_M$ of the material properties of reinforced concrete structures and the uncertainty $K_P$ of the calculation mode of the members can be selected from the reliability design specification for highway engineering [30].

### 2.3. Resistance Probability Modeling

The piers of rigid bridges are mainly thin-walled and of the column type. HLCR bridges usually use single or double, vertical, thin-walled piers, such as the Guizhou–Baishuichuan Bridge, Shaanxi–Shibaochuan River Bridge, Chongqing–Huangshi Bridge, etc. Compared to the single alternative, double thin-walled piers have large comprehensive flexural stiffness, great overall stability, and larger horizontal displacement is allowed. The research object of this paper is a double-limb, thin-walled hollow pier. According to China's regulations [31], when reinforced concrete members are designed for transient conditions, the positive cross-sectional compressive stress at the edge of the concrete in the compression zone should not exceed $0.80f'_{ck}$. Therefore, the resistance of the main pier under the strength failure condition can be written as [30]:

$$R_1 = k_{p1}[\sigma] = 0.80k_{p1}f'_{ck} = 0.80k_{p1}k_{mh}f'_k \tag{16}$$

where $f'_{ck}$ is the standard value of concrete axial compressive strength corresponding to concrete cubic compressive strength $f'_{cu,k}$ during the construction phase (kPa); $k_{mh}$ is the parameter of material property uncertainty in structural members, and its mean value and coefficient of variation can be taken as in [30]; $f'_k$ is the standard value of the material properties of the specimen in the specification (kPa); $k_{p1}$ is the uncertainty factor of the resistance calculation model.

In the construction stage of the lower structure of the bridge, the pier mainly bears the effect of its own gravity and wind load. Then, during the construction of the upper beam structure, the load acting on the pier becomes complex. In addition to its own gravity, it must also bear the construction of the upper structure generated by the self-weight of the beam, hanging basket self-weight and other construction live load. At this time, the pier is a typical eccentric compression member. When the load applied to the structure exceeds its ultimate bearing capacity, the high pier structure will lose its original balance, causing an instability phenomenon.

The critical force at the top of the combined pier is derived from the Rayleigh-Ritz method as [30]:

$$
\begin{gathered}
N_{cr} = k\frac{EI_2}{H^2} - 0.3 k_G q H \\
k = -\frac{8}{3}\sqrt{C_1(n-1)^2 + C_2 n^2 + C_3 n + (C_4 + 31) + \frac{1}{3}[52 + C_5(n-1)]} \\
C_1 = 400B^6 - 900B^5 + 855B^4 \\
C_2 = -405B^3 + 81B^2 \\
C_3 = 1025B^3 - 387B^2 + 72B \\
C_4 = -620B^3 + 306B^2 - 72B \\
C_5 = 160B^3 - 180B^2 + 72B \\
B = H_1/H \\
n = I_1/I_2
\end{gathered}
\tag{17}
$$

where $E$ is the modulus of elasticity of concrete (kPa); $k_G$ is the statistical parameter of constant load indeterminacy; $I_1$ is the bending stiffness of a single pier section (m$^4$); $I_2$ is the double-limb, thin-walled part of the pier flexural stiffness (m$^4$); $H_1$ is the free length of a single part of the pier (m); $H$ is the pier height (m); $B$ is the relative position parameter of the variable section of the bridge pier (the dividing point between the two-armed and one-armed piers); $n$ is the ratio of cross-sectional moments of inertia of a single to a double, thin-walled pier; $q$ is the bridge self-weight set degree (kN/m).

Therefore, the resistance of the bridge pier instability can be expressed as [30]:

$$
R_2 = k_{p2} N_{cr} = k_{p2}\left(k\frac{EI_2}{H^2} - 0.3 k_G q H\right)
\tag{18}
$$

where $k_{p2}$ is the uncertainty factor of the resistance calculation model.

### 2.4. Load Probability Modeling

During the construction of a rigid bridge with hanging baskets, especially in the maximum cantilever condition before closing, wind loads, gravity deviation of cantilever casting, and cradle deflections can affect the stability of the bridge structure. This paper mainly considers the structural constant load, construction live load and wind load applied to the structure.

#### 2.4.1. Structural Constant Load

According to the characteristics of the hanging cradle construction, the structural constant load acting on the main pier in each construction stage mainly includes block gravity, block gravity deviation and block unbalance load. After analysis, the effect of the structural constant load produced at the top (bottom) of the pier is calculated as follows:

(1)  Effect of the action of the cast beam section at the top (bottom) of the pier [1]:

$$
S_{N_G} = k_G p_0 + 2\sum_{i=1}^{n} k_G p_i \qquad S_{M_G} = v_p \sum_{i=1}^{n} k_G p_i x_i
\tag{19}
$$

Here, $S_{N_G}$ is the axial force effect in the axial direction at the bridge pier under the action of cast blocks (kN); $S_{M_G}$ is the along-bridge bending moment effect at the bridge pier under the action of cast blocks (kN· m); $k_G$ is the statistical parameter of constant load indeterminacy (according to the literature [30] it obeys a normal distribution, and is taken as $\mu_{k_G} = 1.0212$, $\delta_{k_G} = 0.0462$); $p_i$ is the weight of the $i$-th block (kN); $v_p$ is the relative deviation coefficient of the self-weight of the beam caused by the construction error on both sides of the cantilever (according to the literature [6], it is taken as $\mu_{v_p} = 0.025$, $\delta_{v_p} = 0.15$); $x_i$ is the distance of the center of gravity of block $i$ from the center of the top of the pier (m).

(2) The effect of deviations caused by different overhanging speeds at both ends of the pier top (bottom) [1] is:

$$S_{N_K} = 2k_G p_K + v'_p k_G p_K \tag{20}$$

$$S_{M_K} = \left(1 - v'_p\right) k_G p_K x_K \tag{21}$$

where $S_{N_K}$ is the axial force effect at piers due to beam section deviation (kN); $S_{M_K}$ is the parallel moment effect at piers due to beam section deviation (kN·m); $p_K$ is the beam section weight (kN); $x_K$ is the distance from the center of the beam section to the center of the top of the pier (m); $v'_p$ is the coefficient of imbalance of the pouring speed at both ends, that is, the ratio of the weight of the beam sections being poured on both sides (according to the literature [1], we assume $\mu_{v_p} = 0.5$, $\delta_{v_p} = 0.075$); $K$ is the beam section number being poured.

(3) The effect of the self-weight of the bridge pier at the bottom of the pier [1] is:

$$S_{N_q} = k_G q H \qquad S_{M_q} = 0 \tag{22}$$

where $S_{N_q}$ is the pier bottom axial force effect caused by the self-weight of the bridge pier (kN); $S_{M_q}$ is the pier bottom bending moment effect caused by the self-weight of the bridge pier (kN·m); $H$ is the bridge pier height (m); $q$ is the average self-weight set of bridge piers (kN/m).

### 2.4.2. Construction Live Load

The overhanging construction live load was noted to have two obvious characteristics [32]. One is that the statistical live loads corresponding to the last three sections are basically linear and do not vary much; the other is that the construction live loads show a certain periodicity as the construction sections advance. Based on these characteristics, the live loads during the construction period are described in the form of segmental functions, and the live load distributions of the last three sections and the remaining beam sections are analyzed separately. The live load model is derived as follows:

$$Q_1 = \begin{cases} 236.75(n+3-K) + 149.175 \\ \qquad\qquad K \geq 3 \\ n = [K-3, K-1] \end{cases} \left(R^2 = 0.9977\right) \tag{23}$$

where $Q_1$ is the average value of the live load during the construction period of the last three sections of the beam; $n$ is the constructed beam section number; $R$ is the correlation coefficient calculated by Pearson's product-moment method, responding to the fitting effect.

$$Q_2 = \begin{cases} -23.1(K-5) + 369.9 \\ \qquad\qquad K \geq 4 \\ \min Q_2 \geq 100 \end{cases} \left(R^2 = 0.9209\right) \tag{24}$$

Here, $Q_2$ is the average value of the live load during the construction period for the remaining girder sections, excluding the last three sections (kN/m).

This leads to the following model for the construction live load effect.

(1) The load effect generated by the construction division live load at the top (bottom) of the bridge pier is [1]:

$$S_{NQ} = S_{NQ2} + S_{NQ1} = k_Q b Q_{20} l_0 + 2k_Q \left( b \sum_{i=1}^{K-4} Q_{2i} l_i + b \sum_{i=K-3}^{K-1} Q_{1i} l_i \right) \tag{25}$$

$$S_{MQ} = S_{MQ2} + S_{MQ1} = v_Q k_Q \left[ b \sum_{i=1}^{K-4} Q_{2i} l_i x_i + b \sum_{i=K-3}^{K-1} Q_{1i} l_i x_i \right] \tag{26}$$

where $S_{NQ}$ is the axial force effect at the bottom of the pier caused by the live load of the construction division (kN); $S_{NQ2}$ is the exclusion of the axial force effect at the bottom of the pier caused by the live load action of the last three remaining girder sections (kN); $S_{NQ1}$ is the axial force effect at the bottom of the pier caused by the live load action in the construction division of the latter three sections of the beam (kN); $S_{MQ}$ is the bending moment effect at the bottom of the pier caused by the live load of the construction division (kN·m); $S_{MQ2}$ is the exclusion of the moment effect at the bottom of the pier caused by the live load action of the remaining beam sections of the last three segments (kN·m); $S_{MQ1}$ is the bending moment effect at the bottom of the pier caused by the live load action of the construction division of the rear three sections of the beam (kN·m); $l_i$ is the length of the $i$-th beam section; $l_0$ is the length of block 0; $k_Q$ is the ratio of the actual acting construction load to the theoretically calculated load value (taken as $\mu_{k_Q} = 1$, $\delta_{k_Q} = 0.144$ [33]); $b$ is the width of the beam section (m); $Q_{2i}$ is the average value of the live load during the construction period for the $i$-th girder section, excluding the last three sections (kN/m); $Q_{20}$ is the average value of the live load during the construction period of block 0 (kN/m); $Q_{1i}$ is the average value of the live load during the construction period of the $i$-th beam part of the last three sections (kN/m); $x_i$ is the distance from the center of the $i$-th beam part to the center of the top of the pier (m); $v_Q$ is the deviation factor of the construction distribution live load (taken as $\mu_{v_Q} = 0.25$, $\delta_{v_Q} = 0.0045$ [34]).

(2) The effect of the gravity of the hanging cradle at the top (bottom) of the pier is [1]:

$$S_{N_g} = 2G \qquad S_{M_g} = 0 \tag{27}$$

where $S_{N_g}$ is the axial force effect at the bridge pier under the action of the hanging basket (kN); $S_{M_g}$ is the bending moment effect at the pier under the action of the hanging basket (and the cantilever construction process on both sides of the selected hanging basket design is the same, so take 0 kN·m); $G$ is the weight of the hanging basket (kN).

When one end of the hanging basket fell, we multiplied the impact factor to 2, meaning $S_{N_g} = 2G$, $S_{M_g} = 2Gl$. Here, $l$ is the distance from the center of the hanging basket to the center of the top of the pier.

### 2.4.3. Wind Load

According to the specification in [35], the basic wind speed probability distribution is selected in this paper to be consistent with the generalized Pareto distribution model, with the following parameter values:

$$f(x) = \frac{1}{\alpha e^{(1-k)y}} \tag{28}$$

$$F(x) = 1 + \frac{1}{e^y} \tag{29}$$

Here, $\alpha$ is the scale parameter with a value of 5.5; $k$ is the shape parameter and its value is 0.58; $\xi$ is the position parameter with a value of 8.27; $y$ is the function describing the relationship between the parameters and can be calculated by the following equation. (The model's standard deviation is 2.35).

$$y = -\frac{1}{k} \ln\left(1 - k\frac{x - \xi}{\alpha}\right) (k \neq 0)$$

The design reference wind speed at the reference height of the bridge or member is determined by a calculation based on the basic wind speed, the value of which is [35]:

$$U_d = k_f k_t k_h W_y \tag{30}$$

where $U_d$ is the design reference wind speed at the reference height of the bridge or member (m/s); $k_f$ is the wind resistance risk factor, which can be selected as 1.02 according to the specification in [35]; $k_t$ is the terrain condition factor, taken as 1 for flat and open terrain; $k_h$ is the surface category conversion and wind speed height correction factor (1.53 for the main girders and 1.51 for the piers); $W_y$ is the fundamental wind speed, which as described earlier, is consistent with the generalized Pareto distribution (m/s).

The design wind speed for the construction phase can be calculated by the following equation:

$$U_{sd} = k_{sf} U_d \tag{31}$$

where $U_{sd}$ is the design wind speed during the construction phase (m/s), and $k_{sf}$ is the risk factor of wind resistance during the construction period, which can be taken as 0.84 according to the specification in [35].

The equivalent static gust wind speed during the construction phase of a bridge or member can be calculated by the following equation:

$$U_{sg} = G_V U_{sd} \tag{32}$$

where $U_{sg}$ is the equivalent static gust wind speed during the construction phase of a bridge or member (m/s), and $G_V$ is the equivalent static gust wind coefficient, which can be taken as 1.25 for the main girders and 1.16 for the piers according to the specification in [35].

The equivalent static gust wind load per unit length of the main girder under a cross-bridge wind is calculated as follows:

$$F_{g1} = \frac{1}{2}\rho U_{sg}^2 C_H D \tag{33}$$

where $F_{g1}$ is the equivalent static gust wind load acting on the unit length of the main beam (kN/m); $\rho$ is the air density (kg/m$^3$), which can be taken as 1.25 kg/m$^3$; $C_H$ is the transverse force coefficient of the main beam, which can be taken as 1.3 according to the specification in [35]; $D$ is the characteristic height of the main beam (m).

The calculation gives $F_{g1} = 2.66 W_y D$.

The equivalent static gust wind loads on the bridge piers, under wind loads in the cis-bridge direction, are calculated as follows:

$$F_{g2} = \frac{1}{2}\rho U_{sg}^2 C_D A_n \tag{34}$$

where $F_{g2}$ is the equivalent static gust wind load acting on a unit length of the bridge pier (kN/m); $\rho$ is the air density (kg/m$^3$), which can be taken as 1.25 kg/m$^3$; $C_D$ is the resistance coefficient of the bridge pier, which can be taken as 1.6 according to the specification in [35]; $A_n$ is the area projected downwind on the unit length of the member (m$^2$/m).

The calculation gives $F_{g2} = 3.00 W_y A_n$.

Therefore, under the cross-bridge wind load on a cantilever construction, the cross-bridge load effect generated at the pier bottom is:

$$S_{NF_{Wh}} = 0 \qquad S_{MF_{Wh}} = \frac{\sum\limits_{i=1}^{N} F_{g1}}{L} A_{Wh} h \tag{35}$$

where $S_{NF_{Wh}}$ is the axial force effect at the bottom of the pier under the wind load in the cross-bridge direction (kN); $S_{MF_{Wh}}$ is the bending moment effect at the bottom of the pier under the wind load in the cross-bridge direction (kN·m); $N$ is the total number of cells, dividing the entire length of the main beam into N cells in 1-m units; $i$ is the unit number; $L$ is the length of the main beam at the maximum cantilever (m); $A_{wh}$ is the cross-sectional

area of the main beam in the cross-bridge direction (m$^2$); $h$ is the height of the cross-sectional center of the main beam from the bottom of the pier (m).

Under the action of wind load on the cantilever construction, the effect of wind load in the direction of the pier bottom is:

$$S_{NF_{Ws}} = 0 \quad S_{MF_{Ws}} = \frac{\sum\limits_{i=1}^{M} F_{g2}}{H} A'_{Wh} h' \tag{36}$$

where $S_{NF_{Ws}}$ is the axial force effect at the bottom of the pier under the wind load in the downstream direction (kN); $S_{MF_{Ws}}$ is the bending moment effect at the bottom of the pier under the wind load in the down-bridge direction (kN·m); $M$ is the total number of cells, dividing the piers into M cells from top to bottom at 1 m; $i$ is the unit number; $H$ is the height of the pier (m); $A'_{wh}$ is the cross-sectional area of the bridge pier in the cis-bridge direction (m$^2$); $h'$ is the height of the center of the bridge pier in the direction of the bridge section from the bottom of the pier (m).

2.4.4. Combination of Effects

Our regulation stipulates that the combination of action effects in the construction phase shall be determined by the calculation needs and the conditions in which the structure is located [36]. According to the provisions of [36] on the combination of the action effect in the construction stage with the construction conditions that may occur during the construction of the hanging basket, the construction stage load is combined according to the following working conditions.

(1) Combination I The last girder section is being poured for the cantilever without unusually high winds. In this case, the effects of wind loads are not considered due to their small size.

    a.    The combination of load effects at the top of the pier is:

$$S_N = S_{N_G} + S_{N_k} + S_{NQ} + S_{N_g} \tag{37}$$

$$S_M = S_{M_G} + S_{M_k} + S_{MQ} + S_{M_g} \tag{38}$$

    b.    The combination of load effects at the bottom of the pier is:

$$S_N = S_{N_G} + S_{N_k} + S_{Nq} + S_{NQ} + S_{N_g} \tag{39}$$

$$S_M = S_{M_G} + S_{M_k} + S_{Mq} + S_{MQ} + S_{M_g} \tag{40}$$

(2) Combination II When high winds occur, the last girder section has been cast, the side spans have not yet been closed, and the structure is in the maximum cantilever state.

    a.    The combination of load effects at the top of the pier is:

$$S_N = S_{N_G} + S_{N_g} \tag{41}$$

$$S_M = S_{M_G} + S_{M_g} \tag{42}$$

    b.    The combination of load effects at the top of the pier is:

$$S_N = S_{N_G} + S_{Nq} + S_{N_g} + S_{NF_{Wh}} + S_{NF_{Ws}} \tag{43}$$

$$S_M = S_{M_G} + S_{Mq} + S_{M_g} + S_{MF_{Wh}} + S_{MF_{Ws}} \tag{44}$$

### 3. Reliability Analysis of the Main Pier of an HLCR Bridge with a Cradle Construction

*3.1. Establishment of Functions*

Research shows that the resistance and effect of structural members during construction are random processes, so the reliability function of different construction stages is not consistent, and the function $Z(t)$ for reliability analysis can be expressed as [35]:

$$Z(t) = R(t) - S(t) \tag{45}$$

Two failure states are considered in this paper, i.e., the strength failure due to the construction load at the bottom of the pier and the instability failure due to the construction load at the top of the pier. The strength failure reliability function under the construction load at the bottom of the pier is:

$$Z = R_1 - \left( \frac{S_N}{A} + \frac{S_M x_0}{I_y} + \frac{S'_M y_0}{I_x} \right) = 0.80 k_{p1} k_{mh} f'_k - \left( \frac{S_N}{A} + \frac{S_M x_0}{I_y} + \frac{S'_M y_0}{I_x} \right) \tag{46}$$

The reliability function for instability failure under the construction load at the top of the pier is:

$$Z = R_2 - S_N = k_{p2} \left( k \frac{EI_2}{H^2} - 0.3 k_G q H \right) - S_N \tag{47}$$

where $x_0$ is the height of the pressure zone of the bridge pier section in the direction of the bridge (m); $y_0$ is the height of the pressure zone in the cross-bridge direction of the bridge pier section (m); $I_x$ is the moment of inertia in the cross-bridge direction of the bridge pier section (m$^4$); $I_y$ is the moment of inertia of the bridge pier section in the cis-bridge direction (m$^4$).

*3.2. Target Reliability Indicators*

At present, there is no provision on the target reliability index for the bridge construction period in China. According to Section 3.3.2 of "*The Uniform Design Standard for Structural Reliability of Highway Engineering*", the target reliability index of highway bridge structures belonging to safety class I of ductile damage is 4.7. Meanwhile, there is a target reliability index of 5.2 for highway bridge structures with safety class I belonging to brittle damage. The Ontario (ONTARIO) Highway Bridge Design Code of Canada "*CAN/CSA-S6-00*" section C4.1 sets the target reliability index of members as 3.5, which is the same as "*AASHTO LRFD Bridge*" section C10.5.5.2.1. The European code "*EN 1990*" section B3.2 limits the target reliability of members to 4-5. Combining these codes, and considering the possibility of a higher accident potential during the cantilever construction phase than during the normal use phase, the target reliability of the bridge structure during the construction phase is temporarily set as 5 in this example.

*3.3. Project Examples*

This special bridge across a river is difficult to construct and has a long construction cycle, with the total design requiring 79,024 tons of steel and 452,000 cubic meters of concrete. The whole bridge is designed in separate bridge sections; the main bridge is arranged as an 87 + 6 × 160 + 87-m continuous rigid bridge. The total span length of the left bridge is L = 3057 m and of the right bridge is L = 3145 m. The layout of some of the bridge is shown in Figure 1. The maximum pile length of the main bridge part is 93 m, the maximum height of the bridge is 137 m and the maximum pier height is 125 m. The main pier spans 6#~12#. Among these, 7#~12# adopts the form of a variable-section hollow pier; the upper 65 m of the variable section takes the form of double limbs laterally, the single-limb width is 6.5 m, the wall thickness of the hollow pier is 0.8 m, a horizontal partition is set every 20 m vertically, the wall thickness of the horizontal partition is 0.8 m and the hollow pier wall thickness is 0.8 m. The lower part of the hollow pier adopts an integral single-box and three-chamber structure, where the wall thickness is again 0.8 m. The longitudinal bridge adopts a variable section form, where the top width is 7.5 m, the

slope rate is 1:80. The pier heights across 7#~12# are 122 m, 125 m, 120.5 m, 116 m, 116.5 m and 108 m, respectively. A pier design drawing is shown in Figure 2. The main bridge is arranged as an 87 + 6 × 160 + 87-m single-box, single-chamber, box-type, variable-section, prestressed, concrete, continuous rigid bridge with a joint length of 1134 m. The height of the box girder and the thickness of the bottom plate are changed parabolically by 1.8 times. The height of the root girder at the center line of the box girder is 10.5 m. The height of the span and end girder is 4 m. The cross-section is a single-box, single-chamber, straight-web box girder. The width of the top plate is 12.25 m, the width of the bottom plate is 6.5 m and the length of the cantilever is 2.875 m. The section of block 0# is 14 m long. The thicknesses of the top plate, bottom plate and web are 0.7 m, 1.8 m and 1.1 m, respectively. The thickness of the top plate of other sections is 0.3 m. The thickness of the bottom plate is 1.8 times greater, parabolically curving from root 1.143 m to mid-span 0.32 m. The thickness of the web of blocks 1#~6# and 13#~18# are 0.80 m and 0.55 m, respectively, with blocks 7#~12# forming a transition section. The main girder is divided into 182 cantilevered sections on both sides of block 0#, with a segmental length of 2 × (6 × 3.5 + 6 × 4 + 6 × 4.5) meters, a side span cast-in-place section of 5.6 m and a 2-m section for both the middle and side span. A structural design drawing of the main bridge box girder is shown in Figure 3. The main bridge is constructed by seven "T" type symmetrical cantilevers cast-in-place on piers 6, 7, 8, 9, 10, 11 and 12, except for the 0# girder section, which is cast in brackets beside the top of the pier. The heaviest suspended section was girder 1# with a weight of 2345.2 kN, and the cast-in-place section of the side span was erected with a beryl truss. The weight of hanging baskets and formwork is calculated as 1200 kN, the maximum deformation is no more than 0.02 m and the maximum allowable deviation of concrete pouring at both ends of the cantilever is no more than 3 m$^3$. The reliability of the main pier in the construction process is analyzed under unfavorable conditions.

### 3.3.1. Load Analysis

1.    Structural constant load

(1)    Self-weight of beam section

The weights of the cantilever on both sides of Pier 8# and the length of each beam section are shown in Table 1.

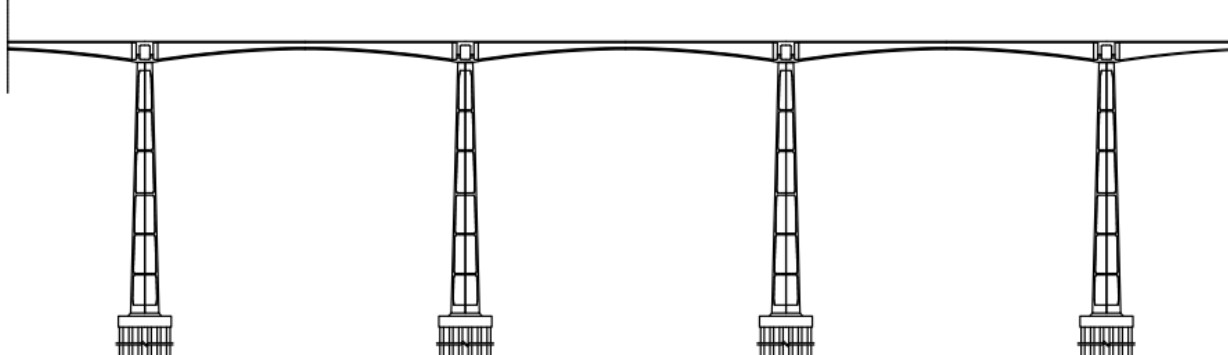

**Figure 1.** Layout of partial bridge.

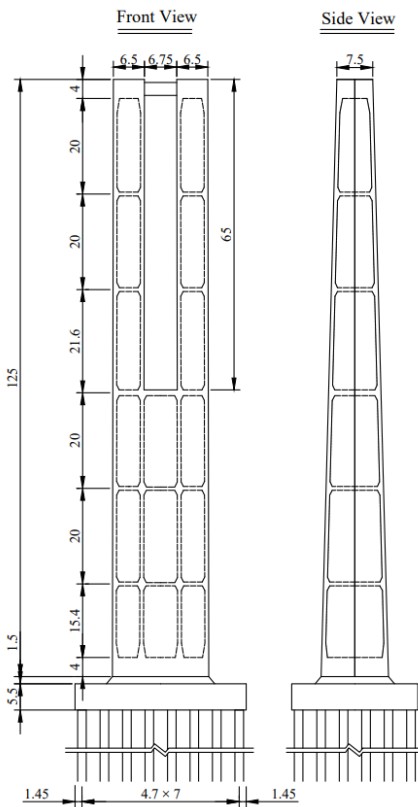

**Figure 2.** Front and side design drawings of Pier 8# (m).

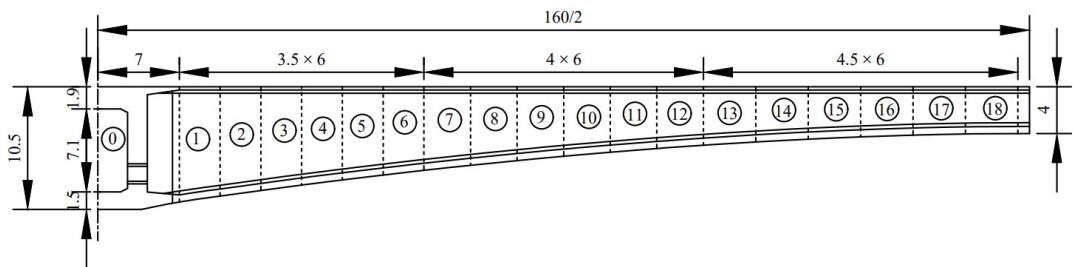

**Figure 3.** General structural drawing of main bridge box girder (m).

**Table 1.** Weight and length of suspended beam section of Pier 8#.

| Beam Section Number | 0 | 1 | 2 | 3 | 4 | 5 | 6 |
|---|---|---|---|---|---|---|---|
| $P_i$(kN) | 16,143.4 | 2345.2 | 2241.2 | 2142.2 | 2046.2 | 1955.2 | 1869.4 |
| $l_i$(m) | 14 | 3.5 | 3.5 | 3.5 | 3.5 | 3.5 | 3.5 |
| Beam section number | 7 | 8 | 9 | 10 | 11 | 12 | 13 |
| $P_i$(kN) | 2035.8 | 1934.4 | 1840.8 | 1755 | 1617.2 | 1432.6 | 1484.6 |
| $l_i$(m) | 4.0 | 4.0 | 4.0 | 4.0 | 4.0 | 4.0 | 4.5 |
| Beam section number | 14 | 15 | 16 | 17 | 18 | | |
| $P_i$(kN) | 1427.4 | 1380.6 | 1341.6 | 1315.6 | 1300 | | |
| $l_i$(m) | 4.5 | 4.5 | 4.5 | 4.5 | 4.5 | | |

From Equation (19), the effect of the cast beam section in the main pier is calculated as:

$$S_{N_G} = k_G p_0 + 2\sum_{i=1}^{n} k_G p_{i\ i} = k_G p_0 + 2 \times \sum_{i=1}^{17} k_G p_i = 73842.2 k_G$$

$$S_{M_G} = v_p \sum_{i=1}^{n} k_G p_i x_i = v_p \sum_{i=1}^{17} k_G p_i x_i = 1175743 v_p k_G$$

(2) Deviation caused by unsynchronized overhanging of beam sections

From Equations (20) and (21), the effect of the deviation of the girder section at the pier due to an unsynchronized suspension speed at the main pier is calculated as:

$$S_{N_k} = 2k_G p_K + v'_p k_G p_K = 2k_G p_{18} + v'_p k_G p_{18} = 2600 k_G + 1300 v'_p k_G$$

$$S_{M_k} = \left(1 - v'_p\right) k_G p_K x_K = \left(1 - v'_p\right) k_G p_{17} x_{17} = 108875 \left(1 - v'_p\right) k_G$$

(3) Self-weight of bridge pier

The self-weight set of the bridge pier is calculated as follows:

$$q = \rho V / H = 26 \times 4526.9 / 125 = 941.60 \text{kN/m}$$

The values of the geometric characteristics of the bridge pier sections are as follows:
Pier top: $I_y = 343.28 \text{ m}^4$, $x_0 = 3.375\text{m}$; $I_x = 457.03 \text{ m}^4$, $y_0 = 3.75 \text{ m}$; $A = 97.5 \text{ m}^2$.
Pier bottom: $I_y = 2967.37 \text{ m}^4$, $x_0 = 6.045 \text{ m}$; $I_x = 255.67 \text{ m}^4$, $y_0 = 5.04 \text{ m}$.
From Equation (22), the effect of the self-gravity of the bridge pier in the main pier body is calculated as:

$$S_{N_q} = k_G q H = 117699.4 k_G$$

2. Live load effect

(1) Construction step-by-step live load

The construction distributed loads on each girder section were calculated according to Equations (23) and (24), as shown in Table 2, and then the construction distributed live loads were calculated according to the length of each girder section in Table 1.

**Table 2.** Construction load on each girder section (Pa).

| No. | 0 | 1 | 2 | 3 | 4 | 5 | 6 |
|---|---|---|---|---|---|---|---|
| Q (kPa) | 0.1 | 0.1 | 0.1 | 0.1 | 0.1 | 0.1 | 0.1 |
| No. | 7 | 8 | 9 | 10 | 11 | 12 | 13 |
| Q (kPa) | 0.1 | 0.1 | 0.1 | 0.1 | 0.1 | 0.1 | 0.1 |
| No. | 14 | 15 | 16 | 17 | 18 | | |
| Q (kPa) | 0.1 | 0.385925 | 0.622675 | 0.859425 | 0 | | |

According to Equations (25) and (26), the load effect of each beam section is first calculated and then combined as the effect of the whole construction distribution live load.

The effect of the construction distributed live load on the bridge pier of the main pier body for girder sections 0 to 14 is:

$$S_{NQ2} = k_Q b Q_{20} l_0 + 2k_Q b \sum_{i=0}^{K-4} Q_{2i} l_i = k_Q b Q_{20} l_0 + 2k_Q Q_2 b \sum_{i=1}^{14} l_i = 149.45 k_Q$$

$$S_{MQ2} = v_Q k_Q b \sum_{i=1}^{K-4} Q_{2i} l_i x_i = v_Q k_Q b \sum_{i=1}^{14} Q_{2i} l_i x_i = 2712.15 v_Q k_Q$$

Then, for girder sections 15 to 17, this is:

$$S_{NQ1} = 2k_Q b \sum_{i=K-3}^{K-1} Q_{1i} l_i = 2k_Q b \sum_{i=15}^{17} Q_{1i} l_i = 205.95 k_Q$$

$$S_{MQ1} = v_Q k_Q b \sum_{i=K-3}^{K-1} Q_{1i} l_i x_i = 2 v_Q k_Q b \sum_{i=15}^{17} Q_{1i} l_i x_i = 7814.83 v_Q k_Q$$

The effect of the distributed live load of the construction of the entire cantilevered girder section on the bridge pier of the main pier body is:

$$S_{NQ} = S_{NQ2} + S_{NQ1} = 355.3998 k_Q$$

$$S_{MQ} = S_{MQ2} + S_{MQ1} = 10526.98 v_Q k_Q$$

(2)　The effect of the self-weight of the hanging basket on the main pier body

This calculation example design of a hanging basket and formwork weight can be taken as $G \approx 1200\text{KN}$.

$$S_{N_g} = 2G = 2400\text{kN}$$

3.　Wind Load

The total lateral area of the main girder of the bridge is $A_{wh} = 535.15 \text{ m}^2$. The height of the cross-sectional center of the main girder from the bottom of the pier is $h = 131.6322$ m. The area of pier 12 in the direction of the bridge is $A'_{wh} = 2048.24 \text{ m}^2$. The height of the center of the pier section from the bottom of the pier in the direction of the bridge is $h' = 101.65$ m.

Therefore, from Equation (35), we can calculate the bending moment effect at the bottom of the pier under the wind load in the cross-bridge direction as:

$$S_{MF_{Wh}} = \frac{\sum_{i=1}^{N} F_{g1}}{L} A_{Wh} h = \frac{\sum_{i=1}^{79} F_{g1}}{79} \times 535.15 \times 131.6322 = 1262.944 W_y$$

Then, calculated by Formula (36) in the downstream wind load, the bending moment effect at the bottom of the pier is:

$$S_{MF_{Ws}} = \frac{\sum_{i=1}^{M} F_{g2}}{H} A'_{Wh} h' = \frac{\sum_{i=1}^{125} F_{g2}}{H} \times 2048.24 \times 101.65 = 10240.0448 W_y$$

3.3.2. Resistance Probability Model

According to Formula (16), the probability of resistance can be derived from the strength of the bridge pier resistance when the damage is:

$$R_1 = 0.80 k_{p1} k_{mh} f'_k = 0.80 \times 32400 k_{p1} k_{mh} = 25920 k_{p1} k_{mh}$$

Calculated from Equation (18), we get $I_1 = 411.0841$, $I_2 = 144.6525$, $n = 2.8419$, $B = 0.48$ and $k = 5.6517$. Therefore, the resistance of the bridge pier at the time of destabilizing damage is:

$$\begin{aligned} R_2 &= k_{p2}\left(k\frac{EI_2}{H^2} - 0.3 k_G qH\right) = k_{p2}\left(5.6517 \times \frac{3.45 \times 10^7 \times 343.28}{125^2} - 0.3 k_G \times 941.60 \times 125\right) \\ &= 4283791 k_{p2} - 35309.82 k_{p2} k_G \end{aligned}$$

3.3.3. Reliability Calculation

Up to now, various load effect and resistance values have been derived. In the following section, we calculate the combined effect of loads under different load conditions. Then, we derive the corresponding reliability function according to Equations (41) and (42), and finally, we calculate the reliability index values. The basic variables of each statistical parameter are shown in Table 3.

**Table 3.** Basic variables for each statistical parameter.

| Name | $k_{p1}$ | $k_{p2}$ | $k_G$ | $k_Q$ | $v_p$ | $v'_p$ | $v_Q$ | $k_{mh}$ | $W_y$ |
|------|------|------|------|------|------|------|------|------|------|
| Distribution Type | Normal | Normal | Normal | Normal | Normal | Normal | Normal | Normal | Generalized Pareto |
| $\mu$ | 1.07 | 1.065 | 1.0212 | 1 | 0.025 | 0.5 | 0.03 | 1.3877 | 8.27 |
| $\delta$ | 0.095 | 0.088 | 0.0462 | 0.144 | 0.15 | 0.15 | 0.15 | 0.1374 | 5.5 |

1.　Combination 1: The last beam section is being poured

　　(1)　The solution for the function of the stability and reliability of the main pier body is:

$$S_N = S_{N_G} + S_{N_k} + S_{NQ} + S_{N_g} = 73842.2k_G + 2600k_G + 1300v'_p k_G + 355.3998k_Q + 2400$$

$$
\begin{aligned}
Z_1 &= k_{p2}\left(k\frac{EI_2}{H^2} - 0.3k_G qH\right) - S_N \\
&= 4283791k_{p2} - 35309.8k_{p2}k_G - 76442.2k_G - 1300v'_p k_G - 355.4k_Q - 2400
\end{aligned}
$$

　　(2)　The solution for the function of the strength of the main pier is:

$$
\begin{aligned}
S_N &= S_{N_G} + S_{N_k} + S_{Nq} + S_{NQ} + S_{N_g} \\
&= 73842.2k_G + 2600k_G + 1300v'_p k_G + 117699.4k_G + 355.3998k_Q + 2400 \\
S_M &= S_{M_G} + S_{M_k} + S_{Mq} + S_{MQ} + S_{M_g} \\
&= 1175743v_p k_G + 108875\left(1 - v'_p\right)k_G + 10526.98v_Q k_Q \\
Z_2 &= 0.80k_{p1}k_{mh}f'_k - \left(\frac{S_N}{A} + \frac{S_M x_0}{I_y}\right) = 25920k_{p1}k_{mh} + 1057.081v'_p k_G \\
&\quad - 3061.61k_G - 11559.425v_p k_G - 3.64512k_Q - 103.497v_Q k_Q - 24.6154
\end{aligned}
$$

2.　Combination II: The last piece is poured and completed to reach the maximum cantilever

　　(1)　The solution for the function of the stability and reliability of the main pier body is:

$$S_N = S_{N_G} + S_{N_g} = 76442.2k_G + 2400$$

$$Z_3 = k_{p2}\left(k\frac{EI_2}{H^2} - 0.3k_G qH\right) - S_N = 4283791k_{p2} - 35309.8k_{p2}k_G - 76442.2k_G - 2400$$

　　(2)　The solution for the function of the strength of the main pier is:

$$
\begin{aligned}
S_N &= S_{N_G} + S_{Nq} + S_{N_g} + S_{NF_{Wh}} + S_{NF_{Ws}} = 76442.2k_G + 117699.4k_G + 2400 \\
S_M &= S_{M_G} + S_{Mq} + S_{M_g} + S_{MF_{Ws}} = 1284618v_p k_G + 10240.0448W_y \\
S'_M &= S_{MF_{Wh}} = 1262.944W_y
\end{aligned}
$$

$$
\begin{aligned}
Z_4 &= 0.80k_{p1}k_{mh}f'_k - \left(\frac{S_N}{A} + \frac{S_M x_0}{I_y} + \frac{S'_M y_0}{I_x}\right) \\
&= 25920k_{p1}k_{mh} - 1991.2k_G - 12629.8v_p k_G - 111.04W_y - 24.6154
\end{aligned}
$$

We can calculate the strength and stability reliability indices under different combinations based on the basic variables of each statistical parameter listed in Table 3 [33] using the JC method with the help of the MATLAB calculation program, to determine that $\beta_1 = 11.1585$, $\beta_2 = 6.7138$, $\beta_3 = 11.1611$ and $\beta_4 = 6.2227$. The derived reliability indicators show that for this bridge, the reliability of the main pier during the construction period is controlled by strength failure and is more dangerous at the time of maximum cantilevering than when the last girder section is poured. This also shows that the wind load magnitude is a key factor affecting the safety of the structure during the construction period.

### 3.3.4. Reliability Parameter Impact Analysis

1. Analysis of the influence of the stability and reliability parameters of the main pier under combination I

Analysis of Figure 4 shows that under this combination, if the coefficient of variation of design variables is higher, the reliability of the structure is lower. Therefore, trying to reduce the variability of the calculation model, materials and loads can improve the reliability of the structure. In addition, it can be seen from the figure that variation of the coefficient of uncertainty $k_{p2}$ for the force-resisting calculation model has the greatest influence on the reliability, followed by the statistical parameter $k_G$ of the structural self-weight. Meanwhile, the variation of the unbalanced parameter $v'_p$ of the casting speed of the beam section and the coefficient of variation of the construction live load $k_Q$ have barely any influence on the reliability of the structure. This indicates that the structural reliability of the bridge constructed by hanging baskets in this combination should be analyzed with the most accurate selection of a resistance calculation model possible, and the control of the construction distributed live load and the synchronous casting speed of the last girder section have little significance for the structural stability and reliability.

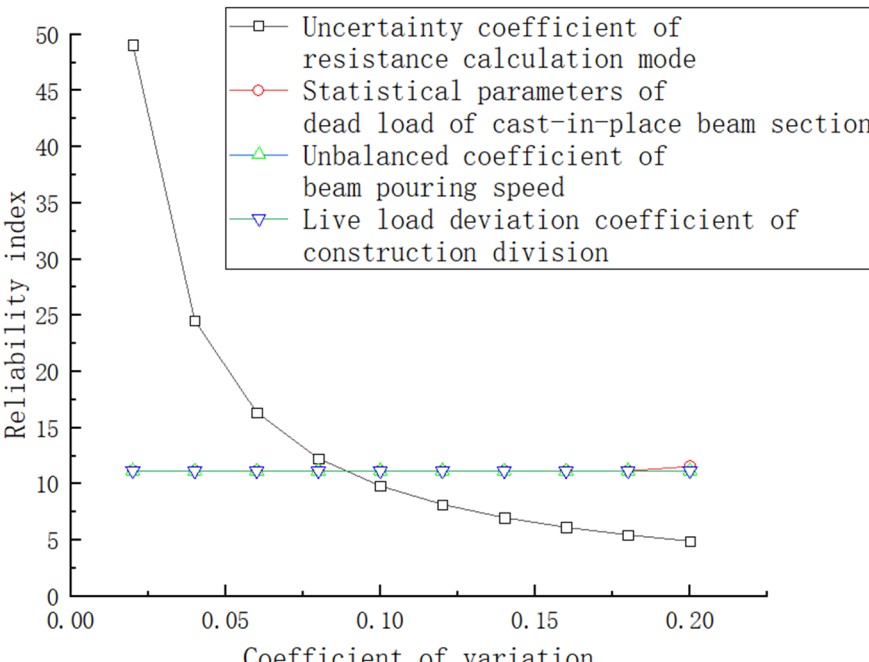

**Figure 4.** Effect of coefficient of variation of each parameter on stability reliability index under combination I.

We can see from Table 4 that under this combined working condition, the change in the mean value of each parameter has little impact on the stability and reliability index.

**Table 4.** Effect of coefficient of average value of each parameter on stability reliability index under combination I.

| $k_{p2}$ | | $k_G$ | | $v'_p$ | | $k_Q$ | | $k_{p2}$ |
|---|---|---|---|---|---|---|---|---|
| Average | Reliability Indicators | Average | Reliability Indicators | Average | Reliability Indicators | Average | Reliability Indicators |
| 1.00 | 11.1451 | 0.90 | 11.1822 | 0.25 | 11.1593 | 0.0 | 11.1594 |
| 1.05 | 11.1555 | 0.95 | 11.1724 | 0.30 | 11.1592 | 0.5 | 11.1589 |
| 1.10 | 11.1650 | 1.00 | 11.1626 | 0.35 | 11.1590 | 1.0 | 11.1585 |
| 1.15 | 11.1731 | 1.05 | 11.1528 | 0.40 | 11.1588 | 1.5 | 11.1580 |
| 1.20 | 11.1816 | 1.10 | 11.1430 | 0.45 | 11.1586 | 2.0 | 11.1575 |
| 1.25 | 11.1889 | 1.15 | 11.1332 | 0.50 | 11.1585 | 2.5 | 11.1571 |
| 1.30 | 11.1956 | 1.20 | 11.233 | 0.55 | 11.1583 | 3.0 | 11.1566 |
| 1.35 | 11.2019 | 1.25 | 11.1135 | 0.60 | 11.1581 | 3.5 | 11.1561 |
| 1.40 | 11.2077 | 1.30 | 11.1036 | 0.65 | 11.1580 | 4.0 | 11.1557 |
| 1.45 | 11.2130 | 1.35 | 11.0937 | 0.70 | 11.1578 | 4.5 | 11.1552 |
| 1.50 | 11.2181 | 1.40 | 11.0838 | 0.75 | 11.1576 | 5.0 | 11.1547 |

2. Analysis of the influence of reliability parameters on the strength of the main pier under combination II

The analysis in Figure 5 shows that the strength reliability decreases with an increase in the coefficient of variation of statistical parameters in this combined condition. Therefore, to improve the strength reliability of this combined structure, variability of the calculation mode, construction material and load deviation should be reduced as much as possible. The variation of the statistical parameter $k_{mh}$ of concrete strength has the greatest influence on the reliability index of the structure, followed by the variation coefficient $k_{p1}$ of the resistance calculation mode, the statistical parameter $k_G$ of self-weight, the coefficient $v'_p$ for imbalance of the block casting and the coefficient $v_p$ for deviation of the cast blocks. The variation of the statistical parameter $k_Q$ and the deviation coefficient $v_Q$ of the construction live load has barely any effect on the structural strength reliability index. This means that to improve the reliability of the structural strength under such combined conditions, it is necessary to strictly control the strength of the concrete materials and the self-weight of the blocks.

We can see from Table 5 that under this working condition, the change to the mean value of each parameter has little impact on the strength reliability index. The difference in reliability index between simultaneous pouring at both ends and pouring at one end before pouring at the other end is not significant, and both approaches meet the requirement of a reliability index measuring 5.

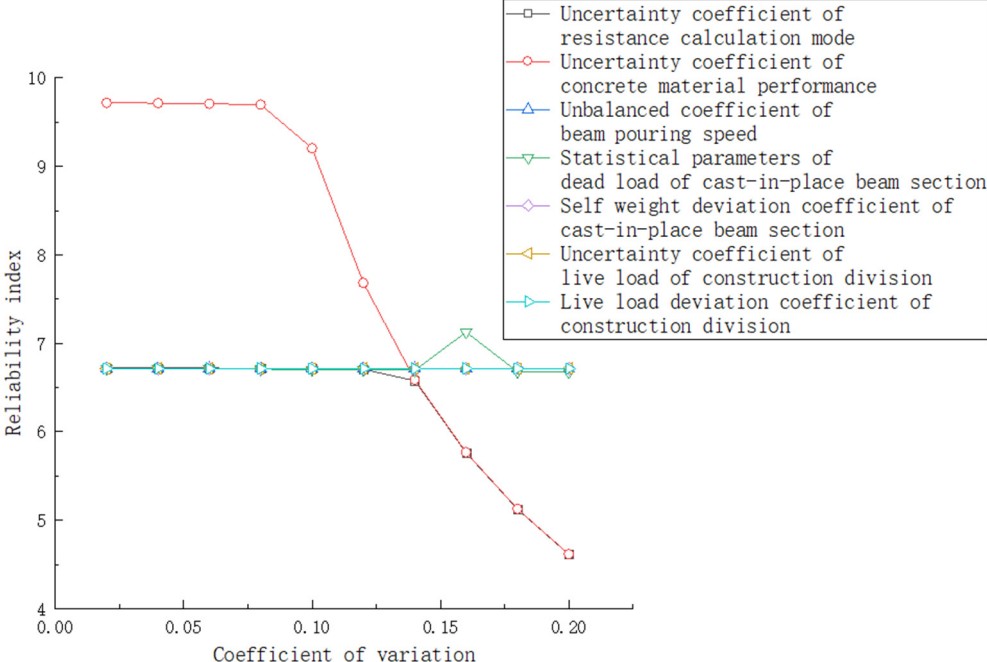

**Figure 5.** Effect of coefficient of variation of each parameter on strength reliability index under combination I.

**Table 5.** Effect of coefficient of average value of each parameter on strength reliability index under combination I.

| $k_{p1}$ | | $k_{mh}$ | | $v'_p$ | | $k_G$ | |
|---|---|---|---|---|---|---|---|
| Average | Reliability Indicators | Average | Reliability Indicators | Average | Reliability Indicators | Average | Reliability Indicators |
| 0.90 | 6.6044 | 1.00 | 6.4882 | 0.25 | 6.6611 | 0.90 | 6.7815 |
| 0.95 | 6.6407 | 1.05 | 6.5269 | 0.30 | 6.6717 | 0.95 | 6.7536 |
| 1.00 | 6.734 | 1.10 | 6.5620 | 0.35 | 6.6823 | 1.00 | 6.7257 |
| 1.05 | 6.7028 | 1.15 | 6.5940 | 0.40 | 6.6928 | 1.05 | 6.6977 |
| 1.10 | 6.7296 | 1.20 | 6.6232 | 0.45 | 6.7033 | 1.10 | 6.6697 |
| 1.15 | 6.7539 | 1.25 | 6.6501 | 0.50 | 6.7138 | 1.15 | 6.6416 |
| 1.20 | 6.7763 | 1.30 | 6.6748 | 0.55 | 6.7243 | 1.20 | 6.6135 |
| 1.25 | 6.7968 | 1.35 | 6.6977 | 0.60 | 6.7348 | 1.25 | 6.5853 |
| 1.30 | 6.8156 | 1.40 | 6.7189 | 0.65 | 6.7452 | 1.30 | 6.5570 |
| 1.35 | 6.8330 | 1.45 | 6.7386 | 0.70 | 6.7556 | 1.35 | 6.5287 |
| 1.40 | 6.8492 | 1.50 | 6.7571 | 0.75 | 6.7659 | 1.40 | 6.5003 |

| $v_p$ | | $k_Q$ | | $v_Q$ | | | |
|---|---|---|---|---|---|---|---|
| Average | Reliability Indicators | Average | Reliability Indicators | Average | Reliability Indicators | | |
| 0.000 | 6.7723 | 0.0 | 6.7152 | 0.00 | 6.7145 | | |
| 0.005 | 6.7607 | 0.5 | 6.7145 | 0.01 | 6.7142 | | |
| 0.010 | 6.7490 | 1.0 | 6.7138 | 0.02 | 6.7140 | | |
| 0.015 | 6.7373 | 1.5 | 6.7132 | 0.03 | 6.7138 | | |
| 0.020 | 6.7256 | 2.0 | 6.7126 | 0.04 | 6.7136 | | |
| 0.025 | 6.7138 | 2.5 | 6.7120 | 0.05 | 6.7133 | | |
| 0.030 | 6.7021 | 3.0 | 6.7114 | 0.06 | 6.7131 | | |
| 0.035 | 6.6903 | 3.5 | 6.7106 | 0.07 | 6.7130 | | |
| 0.040 | 6.6784 | 4.0 | 6.7100 | 0.08 | 6.7128 | | |
| 0.045 | 6.6666 | 4.5 | 6.7093 | 0.09 | 6.7126 | | |
| 0.050 | 6.6547 | 5.0 | 6.7087 | 0.10 | 6.7124 | | |

3. Analysis of the influence of stability reliability parameters of the main pier under combination II

As can be seen from Figure 6, under this combination of conditions, the influence of the uncertainty coefficient of the resistance calculation mode on the structural stability and reliability index is greater, and the influence of the constant load statistical parameter of the cast beam section on the structural stability and reliability is smaller.

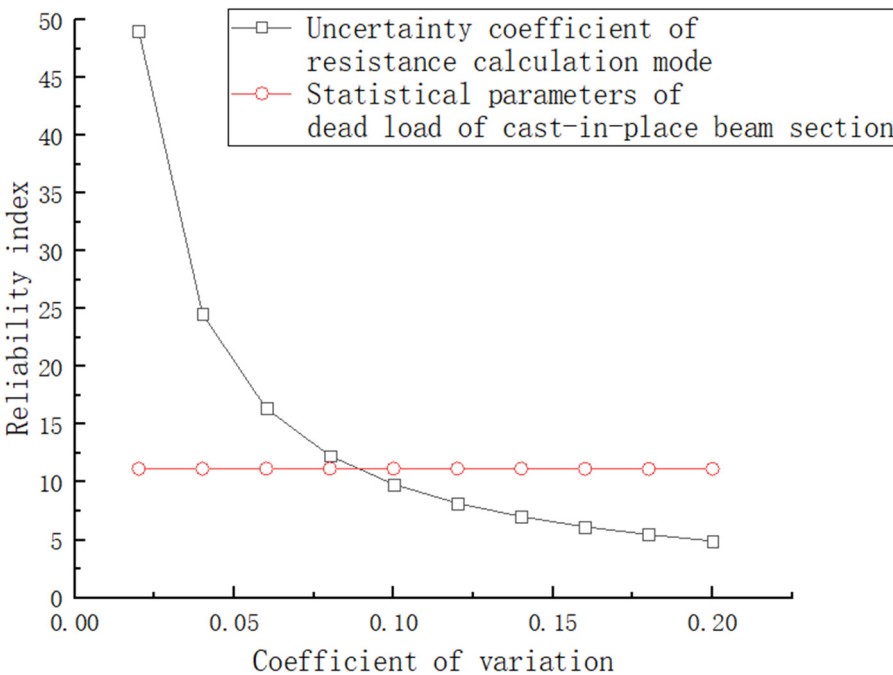

**Figure 6.** Effect of coefficient of variation of each parameter on stability and reliability index under combination II.

Table 6 shows that under this combined working condition, the change in resistance and the average weight ratio of poured blocks have little effect on the change in the reliability index.

**Table 6.** Effect of the average value of each parameter on the stability and reliability index under combination II.

| $k_{p2}$ | | $k_G$ | |
|---|---|---|---|
| Average | Reliability Indicators | Average | Reliability Indicators |
| 1.00 | 11.1479 | 0.90 | 11.1846 |
| 1.01 | 11.1500 | 0.95 | 11.1749 |
| 1.02 | 11.1521 | 1.00 | 11.1652 |
| 1.03 | 11.1542 | 1.05 | 11.1554 |
| 1.04 | 11.1562 | 1.10 | 11.1457 |
| 1.05 | 11.1582 | 1.15 | 11.1360 |
| 1.06 | 11.1601 | 1.20 | 11.1262 |
| 1.07 | 11.1620 | 1.25 | 11.1164 |
| 1.08 | 11.1639 | 1.30 | 11.1066 |
| 1.09 | 11.1657 | 1.35 | 11.0968 |
| 1.10 | 11.1675 | 1.40 | 11.0870 |

4. Analysis of the influence of the strength reliability parameters of the main pier under combination II

As can be seen from Figure 7, the coefficient of variation for the concrete material's property uncertainty has the most drastic effect on the strength reliability index in this condition, followed by the uncertainty of the resistance calculation mode $k_{p1}$ and the statistical parameter $k_G$ of the constant load of the cast beam section, while the effects of $v_p$ and $k_W$ are relatively small.

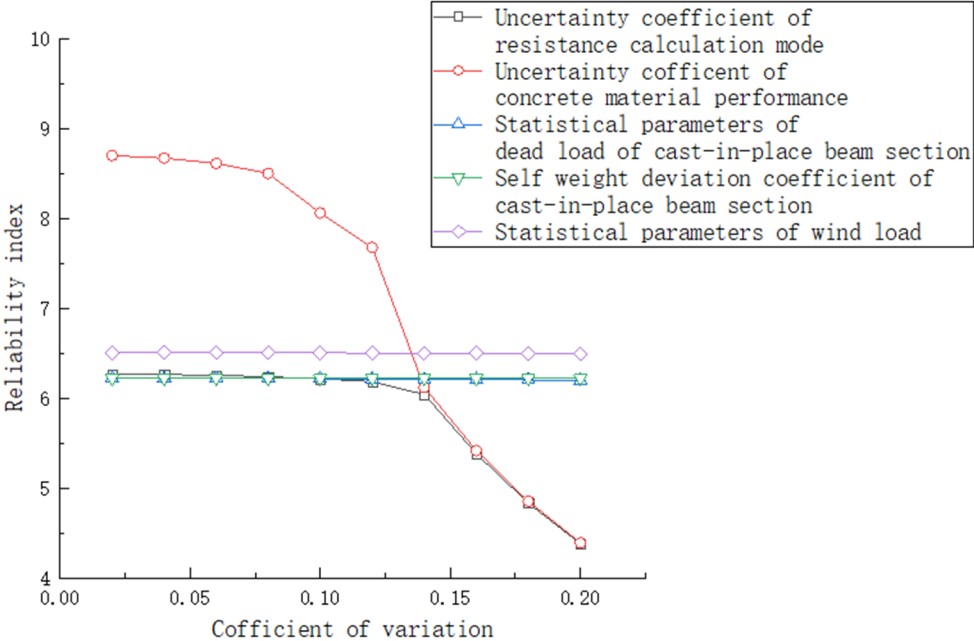

**Figure 7.** Effect of coefficient of variation of each parameter on strength reliability index under combination II.

Table 7 demonstrates that under this combined working condition, the change in the mean value of each parameter has little impact on the strength reliability index, and no matter how the coefficient of each statistical parameter varies, the reliability index meets the requirements of the target reliability.

From the above analysis, it is clear that for this bridge, the reliability of the piers during the cradle construction phase is controlled by the strength failure, and they are more dangerous during the maximum cantilever than when the last girder section is poured. To improve the reliability during the construction period, not only should more consideration be given to improving the structural resistance when designing the bridge but the self-weight error should also be strictly controlled during the construction period to ensure the safety of the bridge project. The construction distributed live load has less influence on the reliability of this bridge failure and thus can be considered less during construction. The variation of wind load parameters has a greater influence on the reliability of the maximum cantilever stage of this bridge, so cantilever cradle construction work should avoid being carried out in windy weather as much as possible.

In addition, during the construction of continuous rigid bridges with hanging cradles, accidents involving a bridge collapse due to falling cradles have occurred. For the bridge in this example, if the hanging cradle falls, the reliability index becomes $\beta_1 = 11.1585$, $\beta_2 = 6.3056$, $\beta_3 = 11.1611$, $\beta_4 = 5.8423$ in order. As can be seen from Table 8, the hanging cradle fall has no effect on the stability reliability, but has some effect on the strength reliability, whereby $\beta_2$ decreases by about 6.08% and $\beta_4$ decreases by about 6.11%.

**Table 7.** Effect of the mean value of each parameter on the strength reliability index under combination II.

| $k_{p1}$ | | $k_{mh}$ | | $k_G$ | |
|---|---|---|---|---|---|
| Average | Reliability Indicators | Average | Reliability Indicators | Average | Reliability Indicators |
| 0.90 | 5.9672 | 1.00 | 5.6882 | 0.90 | 6.2774 |
| 0.95 | 6.0531 | 1.05 | 5.7817 | 0.95 | 6.2548 |
| 1.00 | 6.1294 | 1.10 | 5.8661 | 1.00 | 6.2323 |
| 1.05 | 6.1975 | 1.15 | 5.9425 | 1.05 | 6.2097 |
| 1.10 | 6.2585 | 1.20 | 6.0119 | 1.10 | 6.1871 |
| 1.15 | 6.3135 | 1.25 | 6.0751 | 1.15 | 6.1644 |
| 1.20 | 6.3632 | 1.30 | 6.1328 | 1.20 | 6.1418 |
| 1.25 | 6.4084 | 1.35 | 6.1857 | 1.25 | 6.1191 |
| 1.30 | 6.4495 | 1.40 | 6.2343 | 1.30 | 6.0963 |
| 1.35 | 6.4871 | 1.45 | 6.2790 | 1.35 | 6.0736 |
| 1.40 | 6.5216 | 1.50 | 6.3204 | 1.40 | 6.0509 |

| $v_p$ | | $W_y$ | |
|---|---|---|---|
| Average | Reliability Indicators | Average | Reliability Indicators |
| 0.000 | 6.2859 | 11 | 6.4095 |
| 0.005 | 6.2734 | 12 | 6.3563 |
| 0.010 | 6.2607 | 13 | 6.3007 |
| 0.015 | 6.2481 | 14 | 6.2430 |
| 0.020 | 6.2354 | 15 | 6.1832 |
| 0.025 | 6.2227 | 16 | 6.1214 |
| 0.030 | 6.2100 | 17 | 6.0579 |
| 0.035 | 6.1972 | 18 | 5.9928 |
| 0.040 | 6.1844 | 19 | 5.9262 |
| 0.045 | 6.1716 | 20 | 5.8583 |
| 0.050 | 6.1587 | 21 | 5.7984 |

**Table 8.** Influence of the working condition of the hanging cradle on the reliability index.

| Variables | Hanging Cradles Work Properly | Hanging Cradle Falls |
|---|---|---|
| $\beta_1$ | 11.1585 | 11.1585 |
| $\beta_2$ | 6.7138 | 6.3056 |
| $\beta_3$ | 11.1611 | 11.1611 |
| $\beta_4$ | 6.2227 | 5.8423 |

## 4. Conclusions

In this paper, the existing specifications and related literature were presented. The reliability of the main pier body of an HLCR bridge during cantilever cradle construction was studied. The main conclusions are as follows:

(1) Combined with the design of the HLCR bridge hanging cradle construction and the characteristics of the cantilever construction, the influencing factors of the bearing capacity were determined. The main influencing factors are the uncertainty of the resistance calculation mode, material parameters, the dead load and the live load.

(2) According to the design scheme of the main beam of the HLCR bridge and the technological process of cantilever cradle construction, the structural stress characteristics and failure forms were analyzed in detail. The resistance and action effect models of the main pier under the two most unfavorable conditions were established: (1) There is no abnormal strong wind during the pouring of the last beam section; (2) The last block is poured and reaches the maximum cantilever. At this time, abnormal wind action occurs. Based on these two combinations, a reliability analysis was carried

out, which laid the foundation for reliability analysis of the bridge pier during the construction period.

(3) Based on a river-crossing bridge, the reliability function of a cantilever hanging basket during the construction period was established. We utilized MATLAB software to calculate the reliability index for the main pier during the cantilever construction of the bridge girder by using the JC method. The accuracy of the above analysis was proven, and we verified that the construction process for this project has a sufficient safety margin.

(4) Closely combined with the reliability theory, changes in the statistical parameters of various influencing factors were analyzed for their degree of influence as structural reliability indicators, and it was found that the construction distributed live load has a relatively small influence on the reliability of bridge piers during the construction period. The actual construction can be controlled as little as possible to avoid delaying the construction period. Yet, a change in wind load parameters has a relatively large impact on the reliability of piers at the maximum cantilever, so it is necessary to avoid windy weather during construction. A fall of hanging cradles during construction, increase in resistance, change in the statistical parameters of the structural constant load and the performance of concrete materials all largely affect the reliability of piers and need to be controlled during construction, which provides a reference for the construction of subsequent such projects.

In this paper, our research on the reliability of an HLCR bridge during construction achieved phased results, but there are still areas to be considered and improved. Based on the research process and results of this paper, the following research directions in related fields are proposed:

(1) The large-scale use of finite element analysis software is very beneficial to the study of engineering cases. This paper mainly relied on reviewing the literature when investigating the factors affecting structural reliability during the construction period. Follow-up research could involve finite element software and time-dependent random theory to facilitate a more comprehensive and accurate analysis of the reliability of the bridge structure during construction.

(2) A bridge structure is a whole system composed of many components. The research on bridge structures in this paper was limited to the analysis of structural components, and did not analyze the stability and reliability of a whole bridge system, which should be covered by future research.

(3) The ultimate goal of academic research is to guide production. In this paper, the relative proportions of the factors affecting the reliability during the construction period were determined through analysis. The relative quantitative weight of each factor could be established in future research; such information will be valuable as it can be directly used in the construction of bridges that are yet to be built.

**Author Contributions:** Conceptualization, Q.L.; methodology, T.Z.; validation, Q.L. and T.Z.; formal analysis, N.Q.; investigation, N.Q.; resources, B.X.; data curation, N.Q.; writing—original draft preparation, T.Z.; writing—review and editing, Q.L.; supervision, B.X.; project administration, B.X. All authors have read and agreed to the published version of the manuscript.

**Funding:** This research received no external funding.

**Institutional Review Board Statement:** The study did not require ethical approval.

**Informed Consent Statement:** This study did not involve humans.

**Data Availability Statement:** All the data used in this study can be found in the article.

**Conflicts of Interest:** The authors declare no conflict of interest.

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
