# Peer review of "Reliability Analysis of the Main Pier during the Construction Period of HLCR Bridges"

_applsci, doi:10.3390/app12125936_

Round 1

Reviewer 1 Report

The manuscript focuses on calculations of reliability indices of a typical rigid continuous bridge design. The topic seems interesting, but the overall presentation of the information could be improved. The specific review points are as follows:

1.       The article needs to be thoroughly proofread again to fix word order mistakes, unclear sentences, punctuation, spaces and overall English level. Examples are on lines: 36, 67, 76, 154, 161 and many others. These mistakes also include for example:

a.       The SI prefix “kilo” should be written with lower case k

b.       Section 2.4.3 doesn’t have a title.

c.       Units should be unified, authors sometimes use metres, sometimes centimetres for similar elements etc.

d.       Throughout the article, there are many repeated sentences and information, please thoroughly proofread the text.

e.       The term “functional function” should be written as only “functional” or only “function” depending on the meaning.

2.       All Standards and Codes throughout the article should include a reference to its full name, number, and year. Writing just “the European code” is insufficient.

3.       Section 3.3 could be improved by adding a picture with all the dimensions, pier numbers and other information + relevant cross-section pictures with dimensions, loads etc. The text-only explanation is really confusing and some of the numbers don’t add up.

4.       The first three points in the Conclusions section do not provide any conclusions at all, they are just summaries of the conducted work. Only the fourth point actually mentions some results and provides a very brief conclusion.

5.       Although the Introduction section is very extensive, how is the list of previous studies and methods relevant to this study? Does it build on those previous studies? Or are the previous studies wrong in the author’s opinion and this study provides an improvement?

6.       There is no discussion of the results related to the previous studies mentioned in the Introduction. The authors for example mention that wind load plays an important role, which is the opposite of some conclusions mentioned in the Introduction. Why is that?

7.       It is unclear what is new about this presented study, especially when the authors provide a large list of previous similar studies. It seems that it is rather obvious that concrete quality and wind load are crucial for a large bridge project.

Reviewer 2 Report

The implementation of large structures requires an analysis of their reliability in terms of structural safety. The strength failure and instability damage of the main pier will directly affect the bridge construction safety. The factors that easily affect the stability of the main pier during the construction of high pier and large span continuous rigid frame bridge hanging baskets are in this paper shown . Model fully considers the influence of random factors on the reliability of the pier, and calculates and analyzes the reliability index β through calculation examples. The results show that the changes of various random factors during the construction process have different degrees of influence on the reliability of the bridge pier. It provides a basis for the safety control of hanging baskets in the construction of continuous rigid frame bridges with high piers.

Reliability analysis of the main pier of high pier and large span rigid structure bridge with hanging basket construction was carried out on theoretical grounds. Check Point Method was shown and analysis of indeterminate factors affecting the resistance of main pier in hanging basket construction was performed. Then, resistance probability modeling was performed and load probability modeling.

The presented analysis showed the method of testing this type of structure and the sensitivity level of selected parameters.

The paper has a good scientific level and at the same time points to important elements in the implementation of the structure and should be presented in the Journal of Applied Sciences.

I have no important remarks to this paper, although the Authors are not familiar with the research conducted in this area in various countries.

Reviewer 3 Report

In this paper, the reliability of the main pier in the construction phase of cantilevered hanging baskets for high pier large span continuous rigid frame bridges is studied. The paper is useful for the profession and the academic community, but I noticed certain issues in the paper and I kindly ask the authors to address them.

As far as can be understood from the literature review, most of the references are to works published in China. In addition, since it is not clearly indicated, it can only be assumed that a significant number of papers relate to professional and / or academic project reports.

Using webpages like Scimago Journal & Country Rank it is not possible to find journals Highway Engineering nor Highway. The authors are kindly asked to clarify this.

Also, the authors are kindly asked to indicate with the sources used whether it is a doctoral dissertation or a report, and to point out with the papers written in Chinese (or other language) that the paper is written in Chinese (or other language).

I kindly suggest to the authors to define a simpler and shorter title.

Line 55: "...the construction period at home and abroad." and Line 185: "...domestic and foreign research..." Most of the literature used is writen by Chinese authors. Most of the literature used is domestic. The authors are kindly asked to cite at least five recent papers related to research conducted in Europe and / or the United States.

Line 161-163: Please edit this part of the text. References are messy.

Line 201: "...of the ith..." instead of ith please write i-th.

Line 253: instead of writing "...by the random variable Kp [32].", please write "...by the random variable Kp [32]:". Please make the same corrections throughout the paper.

Line 254-257: "where: fst refers the actual material property values in the structural elements. 

fk refers the standard value of the material properties of the specimen as specified in the specification." Please delete the colon after the word where and make everything as one sentence. Please write as follows: "where fst refers the actual material property values in the structural elements, fk refers the standard value of the material properties of the specimen as specified in the specification." Please make the same corrections throughout the paper.

Line 265: "...span rigid bridges usually use..." Please substantiate the word "usually" with two or three references.

Line 269: "According to China's code,..." Please write specifically which Chinese codes.

Line 273: "where, fck Standards value...". Please write as follows: "where fck is the characteristic value..."

Line 274: "...phase (KPa)." Please use correct SI units. It is kPa, not KPa. Please make the same corrections throughout the paper.

Line 290: "where, E refers modulus of elasticity of concrete (KPa)...." Please write as follows: "where E is modulus of elasticity of concrete (kPa), kG statistical parameter..." Please make the same corrections throughout the paper.

Line 300: "...set degree (KN/m)." Please use correct SI units. It is kN/m, not KN/m. Please make the same corrections throughout the paper.

Line 317: "...cast blocks (KN)." Please use correct SI units. It is kN, not KN. Please make the same corrections throughout the paper.

Line 379: please delete "Error! Reference source not found." and add correct reference.

Expression (37): the plus sign (+) should be written between quotation marks ("+") because it is a combination of loads.

Chapter 3.3 should contain a picture/sketch/draft of the bridge or part of the bridge. The figure will greatly facilitate the interpretation of the text and add dynamism to the manuscript.

Figure 1. Please make the markers on the graph hollow/blank. This will significantly improve the visibility of the results shown in the graph. Please make the same corrections in other similar figures throughout the paper.

Line 776-777: "and the following conclusions are mainly obtained." This sentence sounds strange. Please rephrase. 

Please provide recommendations for further research.

Reviewer 4 Report

The article presents an interesting analysis of the reliability of a complex and responsible structure. The theoretical assumptions for the method of determining the reliability coefficient were presented and, on this basis, calculations were made for a specific case of an existing bridge. The work can be helpful for designers of bridge structures and more. However, this article is more of a case study and does not seem to contribute much to the development of knowledge on the theory of structural reliability. For this reason, I propose to consider the publication of the article after taking into account the following comments by the authors:

1. Authors must demonstrate their own contribution to the development of knowledge on the subject. What's new in their approach to the issue? Do they only present the already existing issues of reliability theory and make calculations? Is it just a case study? What is the innovation of their work?

2. Introduction should be significantly shortened.

3. Conclusions (1) and (2) are not conclusions. Rather, the text is suitable for an abstract or an introduction.

4. The issue presented in the work concerns bridge engineering. The basic language of an engineer are drawings explaining the presented issues. There is not a single sketch or drawing in the paper, which is especially needed in chapter 3. This would significantly improve the readability of the article.

5. The results in Tables 4, 6, 8 and 10 are duplicated in the graphs. Tables should be removed.

6. The text requires a huge number of editorial corrections (punctuation, different fonts and their sizes, etc.)

Round 2

Reviewer 1 Report

The authors extensively improved the manuscript and answered all of my points. I recommend the article to be accepted.

Reviewer 4 Report

I think that after the changes have been made, the article is suitable for publication in an Applied Sciences journal.